# Respiration rate scales inversely with sinking speed of settling marine aggregates

**Kristian Spilling**[1,2]*, **Malte Heinemann**[3], **Mari Vanharanta**[1,4], **Moritz Baumann**[5], **Andrea Noche-Ferreira**[2], **Philipp Suessle**[5], **Ulf Riebesell**[5]

**1** Finnish Environment Institute, Marine and Freshwater Solutions, Helsinki, Finland, **2** Centre for Coastal Research, University of Agder, Kristiansand, Norway, **3** Institute of Geosciences, Kiel University, Kiel, Germany, **4** Tvärminne Zoological Station, University of Helsinki, Helsinki, Finland, **5** Biological Oceanography, GEOMAR Helmholtz Centre for Ocean Research Kiel, Kiel, Germany

* kristian.spilling@environment.fi

**Data Availability Statement:** All relevant data are within the paper and its Supporting Information files.

**Funding:** The mesocosm experiment in 2018 was funded by an EU ERC (Grant No. 695094) and in

## Abstract

Sinking marine aggregates have been studied for a long time to understand their role in carbon sequestration. Traditionally, sinking speed and respiration rates have been treated as independent variables, but two recent papers suggest that there is a connection albeit in contrasting directions. Here we collected recently formed (<2 days old) aggregates from sediment traps mounted underneath mesocosms during two different experiments. The mesocosms were moored off Gran Canaria, Spain (~ 27.9 N; 15.4 E) in a coastal, sub-tropical and oligotrophic ecosystem. We determined the respiration rates of organisms (mainly heterotrophic prokaryotes) attached to aggregates sinking at different velocities. The average respiration rate of fast sinking aggregates (>100 m d$^{-1}$) was 0.12 d$^{-1}$ ± 0.08 d$^{-1}$ (SD). Slower sinking aggregates (<50 m d$^{-1}$) had on average higher ($p <0.001$) and more variable respiration rates (average 0.31 d$^{-1}$ ± 0.16 d$^{-1}$, SD). There was evidence that slower sinking aggregates had higher porosity than fast sinking aggregates, and we hypothesize that higher porosity increase the settlement area for bacteria and the respiration rate. These findings provide insights into the efficiency of the biological carbon pump and help resolve the apparent discrepancy in the recent studies of the correlation between respiration and sinking speed.

## Introduction

Sinking marine aggregates have different characteristics; some are compact and sink quickly whereas others consist of more loosely packed material with lower sinking speed. While sinking, bacteria decompose organic matter, detritivorous zooplankton fragments aggregates and bacterial grazers impose top-down control on remineralization rates [1,2]. Ballasting minerals can affect both the sinking speed and to some extent remineralization rates of the organic material [3,4]. How much carbon is transported to the deep ocean depends among others on the sinking speed and the biological processes connected to the aggregates while sinking.

Traditionally, sinking speed and respiration rates have been treated as independent variables [e.g. 5,6]. Experimental studies using phytoplankton cultures in roller tanks have

2021 by EU H2020 (Grant no 869357), both to UR.
MH was supported by the German Federal Ministry
of Education and Research (BMBF) as Research for
Sustainable Development through the PalMod
project (FKZ: 01LP1505D). Additional funding
came from the AQUACOSM project (EU H2020,
grant 731065) to KS, and the Walter and Andree de
Nottbeck Foundation to KS and MV. The funders
had no role in study design, data collection and
analysis, decision to publish, or preparation of the
manuscript.

**Competing interests:** NO authors have competing
interests.

indicated that the content of ballasting minerals affects sinking speed with respiration rate independent of this at approximately 0.1 d$^{-1}$ [e.g. 6]. However, two recent papers demonstrated that sinking speed affects respiration rates–in contrasting directions. On the one hand, Alcolombri et al. [7] demonstrated that slowly moving aggregates (up to ~40 m d$^{-1}$) are respired faster than barely moving (1–2 m d$^{-1}$) or motionless aggregates, and attributed this to oligomeric breakdown products being more rapidly flushed away at higher flow rates. On the other hand, García-Martín et al. [8] found that respiration rate and bacterial production was higher in the suspended aggregate fraction (assumed to be non-sinking) than in the slow- and fast-sinking fractions (sinking speeds of <24 and >24 m d$^{-1}$).

The microbial communities also differ between aggregates with different sinking speed [9]. Baumas et al [9] demonstrated that fast and suspended aggregates share much of the same community of heterotrophic prokaryotes close to the surface, but at 500 m depth the species richness was much lower and the bacterial production higher in the fast-sinking compared to suspended aggregates. Aggregate size affects the speed but also prokaryotic communities, as large aggregates contain relatively more copiotrophic bacteria than small aggregates [10].

The marine export of carbon to the deep ocean is a major global carbon sink, and much effort has been undertaken to quantify the sinking marine aggregates, which sink from 0 to 1500 m d$^{-1}$ [11–13]. If the respiration rate scales with the sinking speed of settling aggregates, this relationship could be used to refine biogeochemical models to better quantify this biological carbon pump. Here we used sediment collected from two different mesocosm experiments to investigate the relationship between sinking speed and respiration.

## Materials and methods

The material was collected during two different mesocosm experiments off Gran Canaria, Spain in 2018 and 2021. The location was 27.92859 N; 15.36877 E in 2018 and 27.98999 N; 15.36876 E in 2021 (approximately 4 nautical miles apart). This area is coastal, sub-tropical and oligotrophic [e.g. 14].

Sinking material was collected from conical sediment traps attached to the bottom of the mesocosms (Fig 1). The sediment was collected every second day by pumping it out form the sediment trap through a tube going to the surface and into a bottle that was taken back to the laboratory.

In 2018, the overall experimental setup consisted of nine KOSMOS mesocosms [15], which were 15 m deep (Ø = 2 m), each with a volume of 38 m$^3$ [16]. Out of the nine mesocosms, one was an untreated control, and the rest had different addition of deep water, either as a single addition or the same amount of deep-water distributed as multiple additions throughout the experiment [8]. For this study, we used material sampled from three of the mesocosms: the control mesocosm and the two mesocosms that received the highest amounts of nutrient-rich deep-water. Respiration rates were determined on eight different sampling dates (n = 24) [16].

In 2021, the general set-up was also nine, but smaller, mesocosm bags moored next to a pier. Each bag was 8 m$^3$ and the conical bottom reached 4 m deep (the overall shape was similar to the 2018 bags, Fig 1). The experimental set-up was a gradient in alkalinity (from ambient ~2400 µmol kg$^{-1}$ up to ~4800 µmol kg$^{-1}$), obtained by addition of sodium-bicarbonate (NaHCO$_3$) and sodium-carbonate (Na$_2$CO$_3$). In 2021, the collected material from the sediment traps was done only on two different occasions toward the end of the experiment (day 29 and 31), but from all nine bags (n = 18).

We separated the sediment trap material into different sinking speed fractions by using a settling tube. The different sinking fractions were subsequently incubated individually to measure carbon-specific respiration rates. Measurements were done in a temperature-controlled

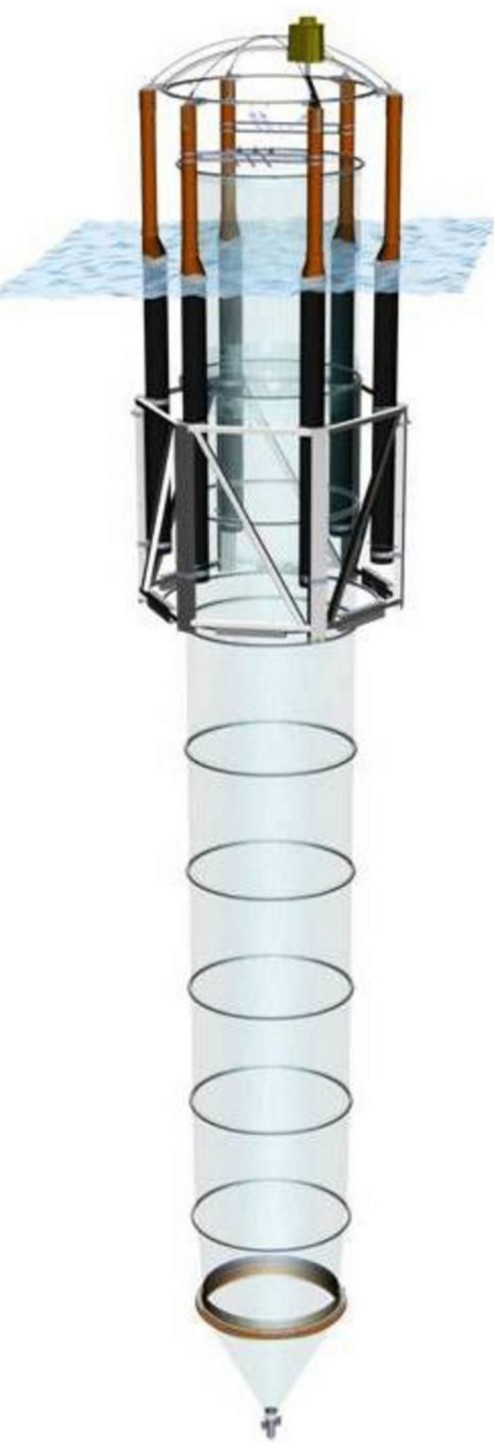

**Fig 1. Schematic drawing of the mesocosm bags used in 2018.** Sinking aggregates were collected from the sediment trap in the bottom at 15 m depth. The material was collected by suction through a tube to the surface every second day. In 2021, we used similar but smaller versions of these bags, with the bottom part reaching 4 m depth. Sampling was done the same way as in 2018.

room at *in situ* temperature (21–23˚ C). The sediment solution (20 ml) was carefully pipetted into the settling tube (Ø = 2.5 cm, total length 100 cm) that was prefilled with filtered (0.2 µm)

seawater. The settling time was 7 minutes, after which the tube was folded in two different locations to divide the settling material into suspended/slow sinking, medium speed and fast sinking aggregates. In 2018, the slow, medium speed and fast sinking fractions were $<10$ m d$^{-1}$, 10–100 m d$^{-1}$ and $>100$ m d$^{-1}$. There were some occasions when there was not enough sediment material in one or more sinking speed fractions to be able to measure respiration during the incubation period (below detection limit, S1 Fig). To prevent this, we did initial tests in 2021 and changed the division to $<50$ m d$^{-1}$, 50–130 m d$^{-1}$ and $>130$ m d$^{-1}$ for slow, medium speed and fast sinking, respectively. This provided a more even distribution of sinking material between the sinking speed fractions and ensured a measurable decrease in $O_2$ concentration during the incubation period in all sinking speed fractions (S2 Fig).

The aggregates from the respective sinking speed fraction were transferred into a 250 ml glass bottle (Schott), which was subsequently filled with 0.2μm filtered seawater, not allowing any headspace. All the bottles were placed in the dark on a rotating wheel (1 rpm) keeping the sediment suspended. $O_2$ sensitive membranes (PreSens, Regensburg, Germany) mounted inside the glass bottles allowed non-intrusive measurements of the oxygen concentration, which was done with a fiber optic cable connected to a Fibox 4 oxygen meter (PreSens, Regensburg, Germany). All of the membranes had been calibrated using 0% and 100% air saturation of $O_2$ following the protocol of the manufacturer: in short 100% by bubbling with air and 0% by addition of sodium dithionite.

Repeated $O_2$ measurements (S1 and S2 Figs) were carried out during the respiration incubation period. These $O_2$ measurements were temperature compensated by the PreSens software, using a thermometer in a 'dummy' bottle containing the same water and being in the same location as the measurement bottles. Three bottles filled with only 0.2 μm filtered water were used as blank controls to see if there was any respiration in the water we added to the sediment. The $O_2$ respiration rate was calculate by linear regression and the data from non-significant (p $>0.05$) slopes were discarded.

After the incubation (24–36 h), all the content of the bottles was filtered onto pre-combusted GFF filters (Whatman). After each filtration, 2 ml 0.5 M HCl was added to remove carbonates from the filters. The filters were dried over night at 60˚C, then packed in clean tin cups and stored dry until determination of the particulate organic carbon (POC) content using a CN analyzer (Euro EA-CN, HEKAtech GmbH, Wegberg, Germany). Respiration of carbon was calculated from the $O_2$ respiration using a respiration quotient of 1 [17]. The start POC was calculated by adding the calculated carbon respired to the measured POC, and the respiration rate d$^{-1}$ was calculated by dividing respired carbon L$^{-1}$ with the start POC concentration (L$^{-1}$). Comparisons between sinking fractions were done with ANOVA on ranks with Dunn's post hoc test for differences between sinking fractions.

## Results and discussion

The respiration rate clearly varied with sinking speed (Fig 2). The average (±SE) respiration rates for both years were 0.42 d$^{-1}$ ± 0.06 d$^{-1}$, 0.24 d$^{-1}$ ± 0.04 d$^{-1}$ and 0.12 d$^{-1}$ ± 0.02 d$^{-1}$ in the slow, medium speed and fast sinking fraction, respectively, and clearly differed between fractions (Dunn's test p $<0.01$; Table 1). Dividing up into the individual years, there was no difference in respiration rate between medium speed and fast sinking fractions in 2018 (Dunn p = 0.29), whereas in 2021 there was no difference between the slow and medium speed (Dunn p = 0.79). The variability in respiration rate was higher in the slow compared with the fast-sinking fraction (Fig 2).

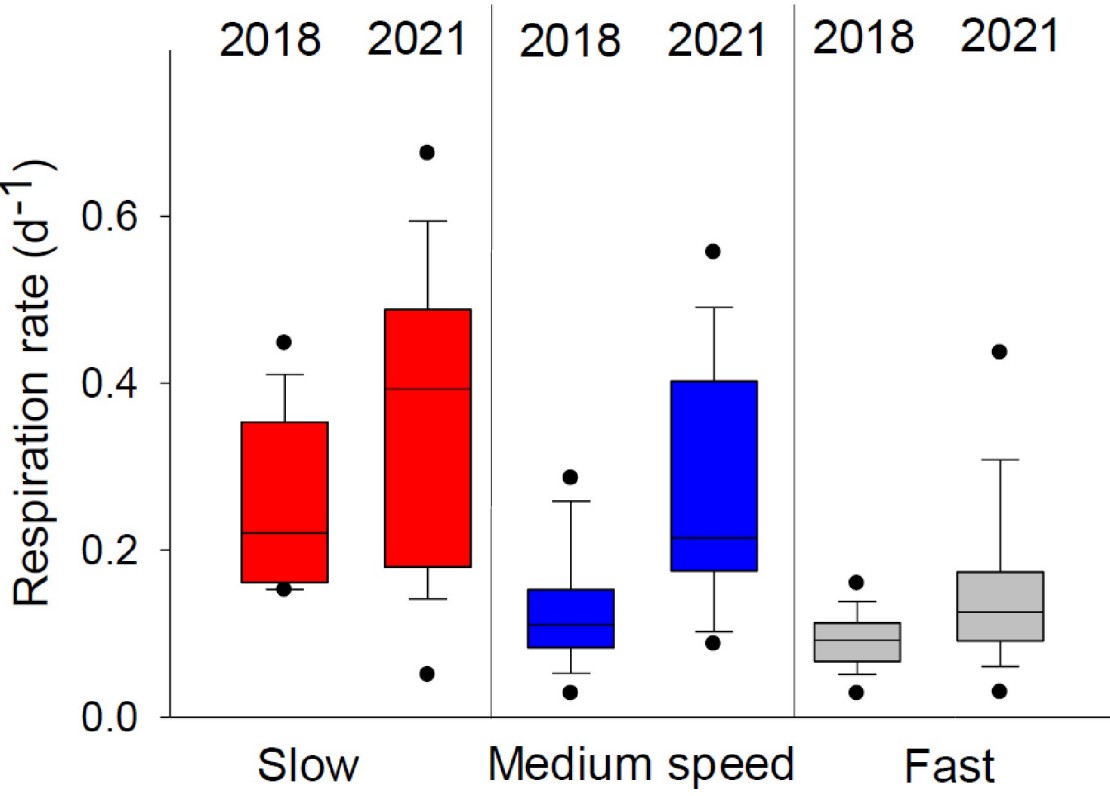

**Fig 2. Respiration rate measured for the three different sinking fractions in 2018 and 2021.** In 2018, this speed division was <10 m d$^{-1}$, 10–100 m d$^{-1}$ and >100 m d$^{-1}$; in 2021 it was changed to <50 m d$^{-1}$, 50–130 m d$^{-1}$ and >130 m d$^{-1}$ for slow, medium speed and fast, respectively. The box outlines the 25–75 percentile, the mid-line is the median, the whiskers are the 10 and 90 percentiles and points are data outside the 10–90 percentile.

There are different characteristics of sinking aggregates that make them sink and decompose at different rates. The contrasting results of García-Martín et al. [8] and Alcolombri et al. [7] for slow-sinking aggregates (0–40 m d$^{-1}$) may depend on the way the measurements were carried out. Alcolombri et al. [7] used uniform aggregates made from agar placed in a flow cuvette, whereas García-Martín et al. [8] collected natural sinking material with a marine snow catcher, and the collected material was subsequently differentiated into different sinking

**Table 1. ANOVA on ranks with Dunn's post hoc test of the different sinking fractions plotted in Fig 1.** The difference of rank means between groups (Diff of Ranks), the Q test statistic (Q) and the p-value (P).

| All data | Diff of Ranks | Q | P |
|---|---|---|---|
| Fast vs Slow | 43.817 | 6.027 | <0.001 |
| Slow vs Medium speed | 21.698 | 2.964 | 0.009 |
| Fast vs Medium speed | 22.119 | 3.089 | 0.006 |
| **2018 data** | | | |
| Fast vs Slow | 26.328 | 5.166 | <0.001 |
| Slow vs Medium speed | 18.104 | 3.506 | 0.001 |
| Fast vs Medium speed | 8.224 | 1.668 | 0.286 |
| **2021 data** | | | |
| Fast vs Slow | 20.667 | 3.941 | <0.001 |
| Slow vs Medium speed | 6.000 | 1.144 | 0.758 |
| Fast vs Medium speed | 14.667 | 2.797 | 0.015 |

fractions. Here we took a similar approach as the latter but with the difference that we collected naturally settling aggregates from mesocosms.

Our results supported García-Martín et al. [8] in that respiration rate was highest in slower sinking aggregates, but note that the division between the sinking speed fractions were not identical as García-Martín et al divided into three groups: suspended, slow sinking (<24 m d-1) and fast sinking (>24 m d-1) aggregates [6]. Our division into different sinking speed fractions was different between the years, which could possibly have shifted the results slightly. However, during both years, the fast sinking particles had lower respiration rate than the slow sinking particles, strengthening the conclusion that sinking speed correlates with the respiration rate.

The increase in respiration with increasing sinking speed observed by Alcolombri et al. [7] was already saturated at 8 m $d^{-1}$, which would not have been picked up with our setup as the slow sinking fraction contained aggregates exceeding this speed. There are *in situ* measurements providing support for the inverse relationship between respiration rate and sinking speed; aggregates in an oligotrophic (Bermuda) location had slower average sinking speed (49 m $d^{-1}$) but higher respiration rates (0.4 $d^{-1}$) compared to a mesotrophic site (Western Antarctic Peninsula; average sinking speed 270 m $d^{-1}$ and respiration rate 0.01 $d^{-1}$) [18]. The warmer water in Bermuda compared to the Antarctic was not sufficient to explain the full magnitude of the difference in respiration between these two sites [18], and our results suggest that different sinking speed could partly explain the difference in respiration rate between these two sites. It is likely not only sinking speed *per se* that affects respiration of marine aggregates, rather properties that affect sinking speed also affect the respiration rate. POC consumption rates also depend on microbial growth dynamics and the ability of bacteria to successfully colonize the particles and overcome multiple loss processes such as viral infection, bacterivory and cell detachment from aggregates [19].

In another recent study, Baumas et al [9] demonstrated that fast sinking aggregates contained less diverse prokaryote communities, but with higher prokaryote production. This would suggest increased carbon remineralization (respiration) in the fast-sinking fraction, but the results are not directly comparable to ours. The production was normalized to cells and the samples were taken much deeper than ours (down to 500 m depth) where presumably the suspended aggregate fraction likely contained mostly refractory organic matter. The inverse relationship between sinking speed and respiration we observed might be depth dependent. All the labile and semi-labile carbon in slow sinking aggregates would likely be consumed, reducing the respiration rate, before these aggregates reach the mesopelagic zone unless they originate from disintegrated fast sinking aggregates.

The sinking aggregates in 2018 were characterized by different aggregate porosities affecting the sinking speed (Fig 3). Slower sinking aggregates tended to have a higher porosity [16], which implies a higher surface to volume ratio and consequently a larger settlement area for bacteria [8]. Porosity has also been found to affect sinking speed [20] and could potentially affect both the sinking speed and the respiration rate. The hydrodynamics around a sinking aggregate, which is affected by speed and shape, controls the initial colonization by bacteria [21]. A positive correlation between water flow and respiration rate caused by removal of degradation products [7] could be overridden by other factors such as surface structure.

The higher variability in respiration rate in the slow sinking aggregates could be due to a more variable aggregate composition than fast sinking aggregates, which are likely more compact, but could still vary in composition [22]. Interestingly, the average respiration rate in the fast-sinking aggregates was similar to what has been obtained from experimental aggregate formation [6], perhaps an indication of suspended and slow sinking aggregates being harder to produce in roller tanks. There are clear indications that community composition is a major

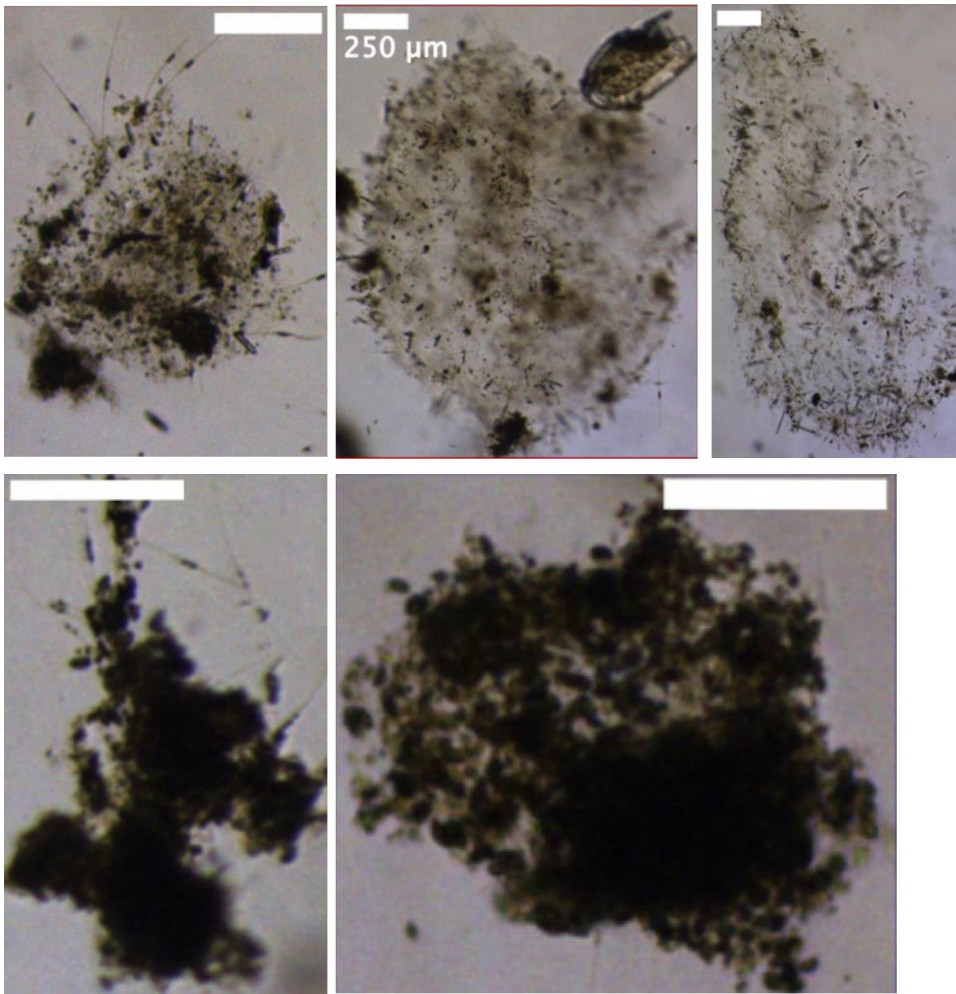

**Fig 3. Example images of aggregates.** High porosity particles in the top row and less porous aggregates in the bottom row. All scale bars are 250 μm. The images are modified from Bauman et al [16] that also provides a more in-depth analysis of how sinking speed relates to porosity.

factor for both sinking speed and respiration [20], but we did not find clear evidence for this in our data.

Here we demonstrated that respiration scales inversely with sinking speed in freshly produced (<2 days old) marine aggregates. There was much higher variability in the respiration rate for slow compared to fast sinking aggregates. The mechanisms behind this variability should be further resolved to better understand the mechanistic drivers of the biological carbon pump in ocean models. There was evidence that increasing porosity decreased the sinking speed [16], and we hypothesize that higher porosity increase the settlement area for bacteria and the respiration rate. Expanding *in situ* imaging technologies could provide better predictions of carbon export in different ecosystems [23].

## Supporting information

**S1 Fig. The raw values of oxygen measured during the respiration period, 2018.** The slow sinking fraction (white), mid sinking fraction (blue) and fast sinking fraction (red). The rows represent different measuring days and the columns the three mesocosms that we extracted

material from (M1, M5 and M8: for the details of the setup see [16]). The x-axis represents time with 00.00.00 being midnight and each minor tick represent one hour. The oxygen respiration was calculated by linear regression from the incubation period and only significant slopes (p < 0.05) were considered. The oxygen respiration was transformed to carbon respired using a respiration quotient of 1, and normalized to the starting particulate organic carbon. (PDF)

**S2 Fig. The raw values of oxygen measured during the respiration period, 2021.** The slow sinking fraction (white), mid sinking fraction (blue) and fast sinking fraction (red). The x-axis represents time with 00.00.00 being midnight and each minor tick represent one hour. The oxygen respiration was calculated by linear regression from the incubation period and all slopes were significant (p < 0.05). The oxygen respiration was transformed to carbon respired using a respiration quotient of 1, and normalized to the starting particulate organic carbon. (PDF)

**S1 Data.**
(XLSX)

## Acknowledgments

We would like to thank the Oceanic Platform of the Canary Islands (PLOCAN) and the Marine Science and Technology Park (PCTM), University of Las Palmas for using their facilities. We also thank the whole KOSMOS team from GEOMAR for logistics and technical help. This study used research infrastructure part of the Finnish Marine Research Infrastructure consortium (FINMARI).

## Author Contributions

**Conceptualization:** Kristian Spilling, Malte Heinemann.

**Formal analysis:** Kristian Spilling, Malte Heinemann, Moritz Baumann.

**Funding acquisition:** Ulf Riebesell.

**Investigation:** Kristian Spilling, Malte Heinemann, Mari Vanharanta, Andrea Noche-Ferreira, Philipp Suessle.

**Methodology:** Kristian Spilling, Malte Heinemann, Ulf Riebesell.

**Project administration:** Ulf Riebesell.

**Supervision:** Kristian Spilling, Ulf Riebesell.

**Writing – original draft:** Kristian Spilling.

**Writing – review & editing:** Malte Heinemann, Mari Vanharanta, Moritz Baumann, Andrea Noche-Ferreira, Philipp Suessle, Ulf Riebesell.

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
