## [Decision Letter · Decision Letter 0]

20 Oct 2022

PONE-D-22-25460Respiration rate scales inversely with sinking speed of settling marine aggregatesPLOS ONE

Dear Dr. Spilling,

Thank you for submitting your manuscript to PLOS ONE. After careful consideration, we feel that it has merit but does not fully meet PLOS ONE’s publication criteria as it currently stands. Therefore, we invite you to submit a revised version of the manuscript that addresses the points raised during the review process.

We look forward to receiving your revised manuscript.

Kind regards,

Amitava Mukherjee, ME, Ph.D.

Academic Editor

PLOS ONE

Journal Requirements:

"We would like to thank the Oceanic Platform of the Canary Islands (PLOCAN) and the Marine Science and Technology Park (PCTM), University of Las Palmas for using their facilities. We also thank the whole KOSMOS team from GEOMAR for logistics and technical help. The mesocosm experiment in 2018 were funded by an EU ERC (Grant No. 695094) and in 2021 by EU H2020 (Grant no 869357). Additional funding came from the AQUACOSM project (EU H2020, grant 731065) and Walter and Andree de Nottbeck Foundation to KS and MV. This study utilized research infrastructure part of the Finnish Marine Research Infrastructure consortium (FINMARI)."

 "The mesocosm experiment in 2018 were funded by an EU ERC (Grant No. 695094) and in 2021 by EU H2020 (Grant no 869357). Additional funding came from the AQUACOSM project (EU H2020, grant 731065) and Walter and Andree de Nottbeck Foundation to KS and MV."

"NO authors have competing interests."

Reviewers' comments:

Reviewer's Responses to Questions

**Comments to the Author**

1. Is the manuscript technically sound, and do the data support the conclusions?

Reviewer #1: Partly

Reviewer #2: Partly

2. Has the statistical analysis been performed appropriately and rigorously? 

Reviewer #1: No

Reviewer #2: Yes

3. Have the authors made all data underlying the findings in their manuscript fully available?

Reviewer #1: Yes

Reviewer #2: No

4. Is the manuscript presented in an intelligible fashion and written in standard English?

Reviewer #1: No

Reviewer #2: Yes

5. Review Comments to the Author

Reviewer #1: I have read and reviewed the manuscript “Respiration rate scales inversely with sinking speed of settling marine aggregates” by Spilling et al.

The authors collect particles in-situ, separated those particles by sinking speed in a settling column, and the performed mesocosm experiments (in which oxygen loss was measured over time) to determine the relationship between size and respiration rate. At this site, on two separate years, the authors showed that fast sinking particles had lower respiration rates than slower sinking particles. However, they showed variability in the respiration rate between pools of faster sinking particles.

Particle remineralization rates and their relationship with particle size are poorly constrained in the literature, and this paper offers an important step forward. However, I had challenges understanding their exact methodology and found some of the writing confusing. I outline my concerns below.

Major comments:

Editing:

The paper needs substantial additional editing – there are numerous grammatical and other writing mistakes. I outline *some* of these in the minor comments. In general, the paper reads like the first author didn’t have enough support by the senior authors in writing. I thin k more eyes on this manuscript would help a lot.

Porosity:

The authors discuss porosity quite a lot, and seem to imply in places that they measure porosity. I don’t see any evidence that porosity was measured.

After re-reading, the authors seemed to imply that smaller particles are more porous than larger ones. I don’t think such a relationship has been established in the literature, and indeed my understanding is that larger particles have a higher fraction of pore space than smaller ones.

Variation in particle remineralization rate:

My understanding is that the authors measured pools of particles, rather than single particles. If I’m wrong here, or even if i’m correct, the authors should clarify. So what is up with the variability in particle remineralization rate. There is lots of variance in replicate runs?

This confusing dialogue around porosity shows up in the abstract and discussion.

See for instance, the observation that particles with increasing diameter increase in mass at rates lower than one would expect from spherical volume alone. See the following reference:

Alldredge AL, Silver MW. Characteristics, dynamics and significance of marine snow. Progress in Oceanography. 1988;20(1):41–82.

Guidi L, Jackson GA, Stemmann L, Miquel JC, Picheral M, Gorsky G. Relationship between particle size distribution and flux in the mesopelagic zone. Deep Sea Research Part I: Oceanographic Research Papers. 2008 Oct;55(10):1364–74.

Abstract

Line 16: Sentence is not grammatical as written.

Its -> Their

Line 18: at -> in

Line 21 0.12/day (add standard deviation)

Line 22: more variable: State the mean and standard deviation

Line 22 Slower sinking particles tened to have a higher porosity: I don’t see where porosity was measured in this paper.

Lines 21 and 22, If the data are really non-normal consider instead reporting median and median adjusted deviation (and maybe range). This approach would be more congruent with figure 1.

22: Smaller particles should have a higher surface aria to volume ratio, even if porosity is held constant. Right? Can the authors be clear that they have deconvolved porosity and size to surface area scaling better here?

Abstract: These processes tend to be location dependent. Please disclose your sampling location in the abstract.

Figures and Tables:

Table 1. Could we ge ta plain language dexcriptionof Diff of Ranks and Q?

Fig 1. What do the circular dots represent? Range?

Supplementary Excel File –

What do the rows represent? My first guess is that each cel is a particle of different sinking speed and its respiration. But then I notice that there are ND columns scattered about in a sort of counterintuitive way. A longer description is clearly in order.

Also what happened when the authors report ND? Do remineralizatio detected? Sample dropped on the floor? Something else?

Introduction

Line 38 – “whereas” implies a contradiction. Ballasting affecting sinking rates is not contradictory with respiration rate of 0.1/day

Introduction: Several papers suggest that ballast minerals protect organic matter from breakdown. I recommend considering, and perhaps citing the following papers

Armstrong RA, Lee C, Hedges JI, Honjo S, Wakeham SG. A new, mechanistic model for organic carbon fluxes in the ocean based on the quantitative association of POC with ballast minerals. Deep Sea Research Part II: Topical Studies in Oceanography. 2001;49(1):219–36.

Lowenstam HA, Weiner S. On Biomineralization. Oxford University Press; 1989.

Mayer LM. Surface area control of organic carbon accumulation in continental shelf sediments. Geochimica et Cosmochimica Acta. 1994 Feb 1;58(4):1271–84.

Methods

Please disclose latitude, longitude of samples collected. and dates of the study.

Describe the oceanography of this site for those less familiar. Such a description could go in the methods or introduction. Is it coastal? Open ocean, how eutrophic/oligotrophic, etc.

I had to read the first paragraph of this methods section about three times to try to figure out what is happening. I think some additional clarity would help . In particular, as someone unfamiliar with these mesocosms, It wasn’t immediately clear to me what a sediment trap attached to a mesocosm is exactly.

In 2018 there are large KOSMOS mesocosms. In 2021 there are smaller mesocosm bags. Both setups have built in, conical, sediment traps on the bottom. Particles are collected form those traps.

A visual schematic overviewing the study design would really help the reader to parse what you did.

Following my comment on the supplementary table, I don’t understand how replicates happen in this experiment and how the size fractions relate to each other.

Line 51: “attached to mesocosms” – I think the authors mean, located inside of the tanks of a mesocosm experiment, which will be summarised elsewhere?

Line 54 – 15 m long. What does long mean here? Deep, or is there a length and width and depth. In which case what is the width and depth?

Line 66 68. “The settling time was seven minutes,” fine. How does this have to do with dividing up the particles by folding the tube. This seems like a sentence with two totally unrelated ideas in it.

Line 69: “Not well distributed” - what does that mean? There wasn’t enough sediment to detect a respiration signal in some fractions?

It looks like the authors redefine slow, medium and fast sinkiing throughout the experiment and then merge these pools in their analysis. They should discuss what such an approach might do to their results.

Line 82: Are the authors normalizing to carbon on the GFF, or normalizing to the sum of carbon on the GFF at the end of the experiment? This seems like too small of a number, because that ignores the carbon that was actually respired right? It seems to me like one would want to estimate the starting organic carbon concentration by summing the GFF measurements with an estimate of total respired carbon, right?

Results and discussion

It is my impression that the first paragraph of this section is results and the rest is discussion. Why not split it into a results section and a discussion section?

The authors collected particles from high nutrient and low nutrient mesocosms. But they dont’ seem to compare the results from the two different environments. Am I missing something, or did this variability just get ignored?

Line 89: Bold the section heading

Line 96: I think you should cite figure 1 here, right?

Line 93-94: No difference -> No difference in respiration rate.

Line 105: Define slow, medium speed and fast particles in the text. Are they the same here as they are for reference 6?

Line 119: Porosity – did you measure porosity in this site, or just size.

120: Sower sinking particles tended to have a higher porosity – I don’t think you measured this. If you are stating something is known, this need s citation.

My experience is the opposite, that large particles have higher porosity.

Discussion overall: I’d like a better picture of why there is so much variability in the sinking speed measurements of the smaller particles.

Reviewer #2: Review of “Respiration rate scales inversely with sinking speed of settling marine aggregates» by Spilling et al.

Comments to Authors:

This manuscript provide interesting new aspects about how respiration rates of prokaryotes attached to sinking particles are correlated with the sinking speed of the particles. The authors convincingly address their main message in a short really clear paper. I appreciate this.

I only have a few major concerns explained below and which could hopefully be addressed easily (mostly in material and method and in discussion).

Major comments:

1)The authors refer along the paper to “respiration rates of sinking particles”, e.g. L19-20 of the abstract among other. I guess you refer to respiration rates of prokaryotes attached to sinking particles or is it to associated zooplankton ? In my opinion, the message would benefit from a precision.

2) Line 48, Despite a good definition and a synthetic review of what is known, I would appreciate one or two sentences more at the end of introduction convincing about why it is crucial to correlate sinking speed and respiration rates?

3) Line 55 to 58 are unclear for me. I don’t get what is the “untreated control” and how adds of nutrients are done and why there. I thought it was a matter of sinking speed and respiration only?

4) Line 67 to 73 : why sinking speed are divided differently between both experiments ? On my point of view it is not crucial for your message however it raises questions. Is that because the decantation times are not the same? Or the height of decantation ? I am just curious.

5) My major concern regarding this paper is about the method to determine respiration rates (line 77 to 86). The manuscript would clearly benefits from additional descriptions here.

I saw that you did blank controls. Is it one bottle dedicated to blank that you used after to remove “blank signal” from the other incubation bottles ? Or did you perform blank in each of the bottle used prior (or after) the experiment ? If I well understand what is written, this was the first option. In that case, did you intercalibrate all bottles by doing 100% - 0% on all sensitive membranes ? In my opinion this is mandatory. If sensitive membranes are still stick inside bottles, it would be easy to check and add it.

In addition, regarding data processing:

- did you verified if respiration data are well linear prior the regression? Could you add (maybe in supp data) residual distribution to check if the model fitted was the best ?

- Was the effect of temperature taken into account ?

- I am really curious about the method you choose to pick the (or one of the) linear part of the curve. This information should be described in the material and method. I would like to have in supp data the raw curve obtained.

6) Regarding the discussion:

- though results are well compared with respiration rates of literature, prokaryotic heterotrophic production (PHP) are related to respiration and paper comparing PHP rates of sinking speed fractions could be relevant (e.g. Church et al. 2021, Baumas et al. 2021 are examples). For instance Church et al. found a higher PHP rate for slow sinking particles than fast sinking. This could confirm your results. Baumas et al. found higher rates for suspended particles than fast sinking per volume of water but the opposite per unic cell of prokaryotes.

Your results are convincing. The rate of respiration is inversely proportional to the sinking speed of the sinking marine aggregates. However, I am a bit frustrated at the end of the discussion. I am curious about how you would explain these inversely relation? Could it be because bacteria are more stressed by the sink when the speed is faster (e.g. pressure increasing, temperature and light decreasing..)? Is it more a matter of porosity as you mention in the introduction ?

What are the implications in term of Bacterial growth efficiency (BGE = PHP/(PHP+PR) with PR the respiration rate), on how marine aggregates are remineralized and thus on sequestration of carbon?

Minor comments:

Sometimes authors refer to “aggregates” and sometimes to “particles”. In my opinion, it is better to choose one and to stay consistent (including title and abstract)

Line 30-31, regarding the topic, it would be interesting and informative to add general sinking speed range with references. Here some examples among other : from 2 to 1500 m d-1 (Alldredge and Silver 1988; Armstrong et al. 2002; Trull et al. 2008). In addition, as my previous comment, I would choose between sinking velocity and speed to help being consistent.

Line 34 : I understand by this sentence that export of Carbon is related only on sinking speed or biological processes. I would add something like “among other” as additional physical, chemical or even composition of particles can act as well (all interconnected finally).

Line 52 : Could it be possible to add a “see after” when referring to respiration rates few paragraphs above its description? I was a bit lost at first but it may be only me…

Line 60 : “sediment was collected” I guess you mean sinking particles which have sedimented and was collected by traps ? Or is it real sediment in the floor of mesocosms? (same question for Line 62)

Line 60 to 61 : “day 29 and 31” Why did you choose these 2 time points ? I guess it is no so important but I am just curious.

I have a question about the rates in “per day”. As it is normalized by POC concentration I guess you mean “by C per day”. Is it mg ? or something else of C per day ? I am not perfectly sure about this

6. PLOS authors have the option to publish the peer review history of their article (what does this mean?). If published, this will include your full peer review and any attached files.

Reviewer #1: No

Reviewer #2: No

---

## [Author Response · Author response to Decision Letter 0]

4 Jan 2023

Please see the 'response to review' file for a detailed response to all the reviewers comments

---

## [Decision Letter · Decision Letter 1]

14 Feb 2023

Respiration rate scales inversely with sinking speed of settling marine aggregates

PONE-D-22-25460R1

Dear Dr. Spilling,

We’re pleased to inform you that your manuscript has been judged scientifically suitable for publication and will be formally accepted for publication once it meets all outstanding technical requirements.

Kind regards,

Amitava Mukherjee, ME, Ph.D.

Academic Editor

PLOS ONE

Additional Editor Comments (optional):

Reviewers' comments:

Reviewer's Responses to Questions

**Comments to the Author**

1. If the authors have adequately addressed your comments raised in a previous round of review and you feel that this manuscript is now acceptable for publication, you may indicate that here to bypass the “Comments to the Author” section, enter your conflict of interest statement in the “Confidential to Editor” section, and submit your "Accept" recommendation.

Reviewer #2: All comments have been addressed

2. Is the manuscript technically sound, and do the data support the conclusions?

Reviewer #2: Yes

3. Has the statistical analysis been performed appropriately and rigorously? 

Reviewer #2: Yes

4. Have the authors made all data underlying the findings in their manuscript fully available?

Reviewer #2: Yes

5. Is the manuscript presented in an intelligible fashion and written in standard English?

Reviewer #2: Yes

6. Review Comments to the Author

Reviewer #2: Good job !

I just spoted a problem with the x axis of supp data figures. Something went probably wrong ?

7. PLOS authors have the option to publish the peer review history of their article (what does this mean?). If published, this will include your full peer review and any attached files.

Reviewer #2: No

---

## [Editor Report · Acceptance letter]

20 Feb 2023

PONE-D-22-25460R1 

Respiration rate scales inversely with sinking speed of settling marine aggregates 

Dear Dr. Spilling:

I'm pleased to inform you that your manuscript has been deemed suitable for publication in PLOS ONE. Congratulations! Your manuscript is now with our production department. 

Kind regards, 

on behalf of

Professor Dr. Amitava Mukherjee 

Academic Editor

PLOS ONE